# Incentive Mechanism and Subsidy Design for Continuous Monitoring of Energy Consumption in Public Buildings (CMECPB): An Overview Based on Evolutionary Game Theory

**Hui Chen [1,\*], Yao Xiao [2], Qiyue Liu [3] and Guanghui Fu [2]**

[1]  Institute of Finance and Public Management, Anhui University of Finance and Economics, Bengbu 233030, China
[2]  School of Economic & Management, Nanjing Tech University, Nanjing 211816, China
[3]  College of Civil Engineering, Nanjing Tech University, Nanjing 211816, China
**\***  Correspondence: chenhui@aufe.edu.cn; Tel.: +86-137-7063-5203

**Abstract:** Rapid urbanization and the continued expansion of buildings have resulted in a consistent rise in the energy consumption of buildings. At the same time, the monitoring of building energy consumption has to achieve the goals of an "Emission peak" and "Carbon neutrality". Numerous energy consumption monitoring systems have been established in several types of public buildings. However, there is a need to ensure that the data are continuously acquired and of superior quality. Scholars have noted that the in-depth research connected to the continuous monitoring of energy consumption in public buildings (CMECPB) is currently sparse. As a result, additional precise quantitative studies targeting the behavior of various stakeholders are also lacking. Hence, there is a need to explore the definition of value and the dynamic benefits of relevant subjects in continuous energy consumption monitoring based on evolutionary game theory and to propose incentive policies. This paper constructs an evolutionary game model for CMECPB between an energy service company (ESCO) and its owner to study the dynamic evolution path of a game system and the evolutionarily stable strategy under market-based mechanisms. Furthermore, by introducing government actions, the incentive policies and subsidy strategy for different subjects of interest are probed in detail by developing a principal-agent model to explore the incentive strength. The following conclusions can be reached: (1) it is inefficient and risky to rely only on the owner and the ESCO in achieving the optimal Pareto equilibrium; (2) the optimal incentives are "fixed incentives" in the case of information symmetry and a "fixed incentive + variable incentive" in the case of information asymmetry; (3) the choice of optimal incentive strategy is also influenced by the cost effort coefficient, risk aversion, external uncertainty, and integrated value transformation coefficient; (4) the incentive intensity and subsidy should be determined by comprehensive analysis with multiple indicators based on the conventional value of a project and the external value of a particular project. An in-depth understanding of each component of the CMECPB pathway yields insights into overcoming the challenges of building energy saving. Furthermore, the results may be useful in developing targeted, effective incentive policies for different disciplines and promoting the continued progress of monitoring building energy consumption and building energy efficiency.

**Keywords:** building energy consumption monitoring; evolutionary game theory; incentive mechanism; subsidy design

## 1. Introduction

Due to rapid urbanization and the rising number of buildings, the demand for building energy continues to increase [1]. The construction industry is responsible for consuming large amounts of energy and ranks among the top three energy-consuming sectors along with industry and transportation. This industry sector accounts for roughly one-third of global energy expenditure and around 40 percent of global $CO_2$ emissions [2]. Over

the lifetime of a building, the majority of its total energy consumption is from the run-in phase [3]. According to the 2022 Global Status Report for Buildings and Construction published by Global Alliance for Buildings and Construction (Global ABC), the building energy demand increased by approximately 4% since 2020 to 135 EJ, the largest increase in the last 10 years. This means that $CO_2$ emissions from building operations are at an all-time high of about 10 billion tons of $CO_2$, about 5% more than in 2020 and 2% higher than the previous peak in 2019. Statistically, large public buildings consume high amounts of energy, and the area of large public buildings accounts for less than 4% of the building area in China, but their energy consumption accounts for more than 20% of the total building energy consumption. The International Energy Agency (IEA) released the World Energy Outlook 2022, which reported that the energy supply crisis has had a negative impact on global economic activity. For countries that rely on energy imports, higher energy prices have raised production costs for businesses, leading to lower economic output, while some countries engaged in energy exports have benefited from higher prices. Thus, it is essential to monitor the energy consumption of buildings and diagnose and improve their energy efficiency by obtaining temporal data of the run-in phase in order to quantitatively characterize the energy consumption of the building facilities. Currently, various energy consumption monitoring systems are used in several types of public buildings; however, factors such as data loss, transient anomalies, and non-continuous monitoring significantly reduce the usefulness of the data [4]. Therefore, it is crucial to continuously monitor the energy consumption of buildings and guarantee consistent data acquisition and quality. At the same time, the availability, validity, and stability of building energy consumption data [5] have continuously been improved by the rapid proliferation and advancement of sub-metering technologies. When relying on specific hardware and software, the precision monitoring of building energy consumption requires the continuous operation of sub-metering devices to ensure that the provided data information is continuous, accurate, and complete. In addition, the owner or user is the main driver of energy consumption in the building [6] and is responsible for regularly troubleshooting the equipment and maintaining the proper operation of the sub-metering equipment. It is essential to effectively operate an energy consumption monitoring platform, improve energy efficiency, and adopt corresponding energy conservation and emission reduction measures.

Most of the current research on building energy consumption focuses on the design of building energy management systems, energy forecasting, and the integration of energy consumption monitoring data. Nowadays, IoT technologies are considered one of the most significant industries, and their use in construction industry applications is becoming increasingly common. They are mainly used in smart buildings, building energy management, and building energy systems to achieve energy-saving and carbon reduction goals [7]. With the growing amount of data on building energy use, new methods for predicting building energy consumption have been explored. Flexible and accurate data-driven methods such as Artificial Neural Network (ANN), Support Vector Machines (SVM), and alternative approaches have been continuously explored [8]. The effective and accurate prediction of building energy consumption plays a key role in implementing accurate control strategies to improve energy use efficiency. Some previous studies have confirmed that there is a significant discrepancy between actual and projected building energy consumption [9], with deviations of more than 30 percent [10]. In addition, it has been shown that building energy consumption is influenced by numerous factors, such as occupant behavior [11], lighting systems [12], etc. Therefore, the accurate prediction of building energy consumption may seem like a daunting task, given the complexity of the relationships among the influencing factors and the diversity of building types [13].

To date, research on building energy consumption monitoring has mainly focused on energy efficiency and the development of emission reduction-oriented tools [14]; however, few studies have been performed on sub-metering devices for continuous data acquisition from energy consumption monitoring systems. Moreover, while focusing on the

influence of incentives on energy behavior [15], few quantitative studies have addressed the incentives of different stakeholders in data acquisition. Some scholars have argued that understanding and extracting the vast amount of data available on building energy consumption are essential for the development of energy-saving incentive policies for buildings [16]. However, we should note that the current research on building energy consumption monitoring ignores numerous additional crucial issues, such as the impact of government actions on various disciplines. In the absence of an insightful analysis of these issues, the effectiveness of building energy consumption monitoring can be hindered, and the development and exploration of incentives can be fairly challenging. In addition, numerous scholars have conducted extensive research on incentive policies for both governments [17] and developers. Nevertheless, the research on incentive policies using tripartite game analysis methods has not been sufficiently thorough, especially regarding the lack of multi-view and multi-type incentive policies.

The present study seeks to fill the above deficiency. We aim to explore the definition of value for subjects related to CMECPB based on game analysis, which is valuable in developing policies for building energy conservation. Based on the analysis of the problems and causes of CMECPB, we construct an incentive model and mechanism considering the results of the analysis. This paper aims to answer the following scientific questions: Do the subsidy incentives provided by the government work, and who are the most efficient? How much subsidy is appropriate? Do the subsidy incentives provided by the government work, and which ones are the most efficient? How much subsidy is appropriate? The main innovation of the proposed approach is to construct an optimal incentive model based on the principal-agent theory between the government, the owner, and the energy-saving service company from an ideal perspective. By introducing government actions, the market balance of energy consumption CMECPB in public buildings can be achieved through effective recommendations from policy, management, measures, and technology. Valuable results from previous research on building energy forecasting can provide a foundation and more targeted guidance for the implementation and promotion of incentive policies in the context of building energy monitoring. By combining continuous monitoring utility analysis with evolutionary game results, we define the integrated value of CMECPB. Moreover, we explore the implementation barriers of incentive mechanisms, which will help to significantly explore more reasonable incentives for CMECPB and guide the government to formulate and implement more effective policies.

## 2. Literature Review

### 2.1. Building Energy Efficiency and Energy Conservation Systems

Buildings for various purposes, such as residential, educational, office, medical and industrial, are emerging as key energy consumers. Energy consumption in buildings involves various subsystems such as lighting, domestic and commercial appliances, as well as Heating Ventilation Air Conditioning (HVAC) systems [18,19]. In this context, the current work focuses directly on meeting building energy efficiency regulations by assuring operational requirements under essentially imaginable energy costs and environmentally friendly conditions [20]. Energy consumption by built environments can be reduced through new designs, technologies, and materials; proper control; and the use of effective energy management systems by considering factors such as building orientation, shape, wall–window ratio, insulation, use of high-efficiency windows, and natural ventilation [21]. In any case, improving the energy efficiency of these subsystems is a test of people, as they generally have to meet complex work specifications, dynamic energy requirements, and comfort standards [22]. For example, several energy-saving measures were proposed to study the cost optimization analysis of energy-efficient residential buildings and were applied to different parts of the buildings [23]. Moreover, by collecting energy data before and after the transformation, energy use models can be used to calculate the energy savings of various energy conservation measures [24]. These studies provide an influential basis for figurative measures of building energy saving.

Data shows that energy consumption and greenhouse gas emissions in the building sector are growing at an advanced rate than in other sectors [25]. A number of studies that analyze the importance of energy efficiency and green buildings exist in the international literature [26]. In order to meet the requirements of building energy efficiency, the concept of a Building Energy Management System (BEMS) is being used to save energy and achieve a sustainable society for new and existing buildings worldwide [27]. A BEMS, which is a heterogeneous system, provides space-based real-time energy monitoring [28], which can be described as a combination of strategies and methods needed to improve the performance, efficiency, and energy use of a building [29]. The primary objective of a BEMS is to improve user satisfaction and reduce energy consumption, operating costs, and environmental impacts, including carbon emissions, by creating a comfortable and efficient environment. In addition, a BEMS is a highly complex information-gathering and control system and implements advanced control strategies [14], as it has monitoring and communication facilities to save staff time and provide commissioning benefits [30]. Therefore, BEMSs are considered a critical tool to achieve energy efficiency [31].

BEMSs with different energy usage models have been designed to optimize energy cost and performance under conditions that do not compromise occupant and user comfort [32]. Missaoui et al. [33] explored the Global Model Based Anticipative Building Energy Management System (GMBA-BEMS), an expected building energy management system based on a global model, which takes into account both user expectations and physical constraints such as energy prices and power constraints and aims to compromise the optimization of user comfort and energy costs. Rocha et al. [34] simulated a smart BEMS with dynamic temperature settings, incorporating heating and cooling system operation decisions and energy source decisions to construct an integrated model. It has been shown that smart BEMS can save more energy than conventional BEMS [35].

Naturally, systems built to achieve overall building energy efficiency are not limited to BEMS. Energy Consumption monitoring technology also can effectively reduce the energy consumption of public buildings in the operational phase [36]. Soudari et al. [37] proposed personalized energy management systems that can specify comfort in a rolling time domain, innovatively encompassing demand response, occupancy and occupant behavior. Energy management and information systems provide meter-level anomaly detection and diagnosis solutions, which give rapid insight into anomalous and non-anomalies energy performance patterns in buildings and reduce energy waste by enhancing the effectiveness of decision-making [38]. Kim and Ha [39] suggested the effectiveness of IoT systems and BEMS in supplying the building users and management with high-resolution, low-cost data acquisition systems highlighting the existing challenges and future scopes. In addition, applications in areas related to energy management subsystems are expanding owing to more detailed studies on improving the energy efficiency of building energy subsystems. The Smart LED Lighting and Efficient Data Management System, which combines lighting and building management systems, not only improves visual comfort but also increases energy utilization [40]. Reducing the energy consumption and carbon emissions of buildings has become a hot research topic in the construction field as research deepens, and it continues to be corroborated by pointers.

### 2.2. Evolutionary Game Theory

Game theory, which focuses on interactions between participants, proposes hypotheses for their behavior, and predicts the outcome, has been a hot topic in the scientific community [41]. It uses mathematical models to foresee the behavior of players in situations of cooperation and conflict [42] and is also commonly applied to environmental policy [43,44] and population evolution [45]. Based on game theory, this study area forms evolutionary game theory, which has been developed mainly by biologists [46]. The main difference between the two is whether the evolutionary process is dynamic or not [47]. While traditional game theory focuses on strategic decision-making and assumes absolute rationality on the part of the players, evolutionary game theory assumes finite rationality

on the part of the players, reaches game equilibrium through trial and error, and pays extra attention to the dynamical process of the game [48]. Evolutionary game theory is also not limited to dynamics and equilibrium, as it can also provide participants with lessons from previous experience [46]. In light of this, scientists have focused on constructing two-party or three-party evolutionary game models in the field of building energy-saving or green buildings. To explore the decision-making mechanism in the dynamic game process, Fan and Hui [49] developed a two-party game model between the government and developers, which promotes policymakers to predict developer behavior and maintain long-term incentives by quantifying returns and simulating interactions. In the study of tripartite games, Jiang et al. [50] introduced trade associations and idle penalties into the constructed tripartite evolutionary game model of the government, developers, and homebuyers. Their research showed that key factors such as punishment, subsidy, and cost have a significant influence on the evolution trend of the dynamic system between the three parties and that they can promote the development of green residential buildings more effectively. Liu et al. [51] focused on the supply market of green buildings, established a tripartite evolutionary game model for governments, suppliers, and developers and stated that tripartite green behavior and strategies cannot leave the public oversight mechanism. Yin et al. [52] developed two three-way evolutionary game models from the supply-side and demand-side perspectives of green buildings to study the multi-stage governance mechanism in the transition process.

Based on the above, researchers have introduced evolutionary game models for building energy fields. Relevant studies focused on discussions between governments and property owners while seeking ways to improve energy efficiency in construction. For instance, Yang et al. [53] revealed the game strategy changes of encouraging green retrofits and implementing green retrofits in government groups and investment groups through an evolutionary game analysis. Research shows that the increased benefits and reduced costs have the best incentive effectiveness. The positive policy incentive measure will reduce the number of investment groups that implement green retrofits, while the negative policy incentive measure will achieve a high level of effectiveness. Mahmoudi et al. [54] used a two-group evolutionary game approach to model and analyze the comparison of government and producer goals in different scenarios. Their results showed that government actions have a strong impact on the activity of producers and competitive markets. Zhang et al. [55] conducted an evolutionary game analysis between developers and the government and proposed that the government should increase the incentives to promote the development of green retrofitting of large public buildings. Zhou et al. [56] used evolutionary game theory to build green technology innovation activities of the government, public, polluting enterprises, and non-polluting enterprises of a four-group evolutionary game model under environmental regulations to discuss and analyze the strategic stability of each game subject and the influence mechanism of strategic selections.

In addition, evolutionary game theory is also widely used in logistics network systems, supply chains, and so on. Mohammad Asghari et al. [57] investigated a dynamic mathematical model to evaluate the incentive effect on the return quantity of used products in a reverse logistics network design, which included different stages of managing waste product distribution through the coordination between collection centers and recovery facilities to warrant economic efficiency. The innovation lies in a fuzzy approach applied in the optimization model to interpret the relationship between the rate of return and the suggested incentives. Wan et al. [58] established a cooperative poverty alleviation model with multi-subject participation composed of a cooperative, smart supply chain platform and the government. The evolutionary game method was used to explore the cooperative poverty alleviation strategy of smart supply chain platforms and cooperatives under the government financial platform subsidy mechanism. In particular, the game equilibrium of the smart supply chain platform and cooperative in cooperation with and without the government financial platform subsidy was analyzed. This paper applies game theory to the monitoring of energy consumption in public buildings. The innovation is that this

paper considers the incentive target, value requirement, and the influencing factors of the optimal incentive according to the results of the evolutionary game of stakeholders, and by constructing an incentive model, the incentive mechanism is designed in a more targeted way, which provides an institutional guarantee for the government side to promote the energy consumption monitoring of public buildings.

Indeed, several studies have found that evolutionary game theory is well-placed to study the variation and stability of strategies among players in a building domain, which is the basis of the model applied in our study. Building on these developments, this paper seeks to contribute to constructing an evolutionary game model of the owner, ESCO, and the government, in which the owner and ESCO are considered key players in CMECPB, and the government is the promoter and guide of the building energy efficiency market, promoting energy consumption monitoring through certain incentive policies. Through the game analysis of multiple stakeholders, the mutual influence and evolutionary trends of the parties are discussed, and references and suggestions for energy consumption monitoring are provided.

## 3. Model Building
### 3.1. Research Design and Model Usability Analysis

The whole research framework of this paper is clear. Based on the full analysis of the background and the summary of the research status, the qualitative and quantitative analysis is carried out for the difficulty of collecting energy consumption monitoring data. The evolutionary game model and the principal-agent model are used to analyze the problem of whether to motivate and how much to motivate. Finally, a solution to the difficulty of collecting energy consumption monitoring data is proposed, an incentive mechanism is designed, and corresponding incentive policy recommendations are given. The technical roadmap of this paper is shown in Figure 1.

The incentives for CMECPB are designed to address the lack of incentives for continuous monitoring and to promote the marketization of building energy efficiency, which will alleviate and reduce the problem of inconsistent data used for building energy consumption decision-making. Building energy saving also has strong positive externalities, that is, building energy conservation can bring additional social and environmental benefits. Therefore, energy conservation and environmental protection will be the main themes in the future development of the building energy conservation market, which in turn will promote the continuous renewal and improvement of incentive means. Evolutionary game models based on game analysis provide a framework for elaborating incentive policies for different disciplines by finding the best perspective on the general and external value of a project to the government, the owners, and the ESCOs. These models appear to be valuable in fully demonstrating where the main potential for technology-related continuous monitoring lies in the relevant sector of the continuous monitoring of building energy consumption, which is useful for identifying practical constraints that can be expected from policy measures.

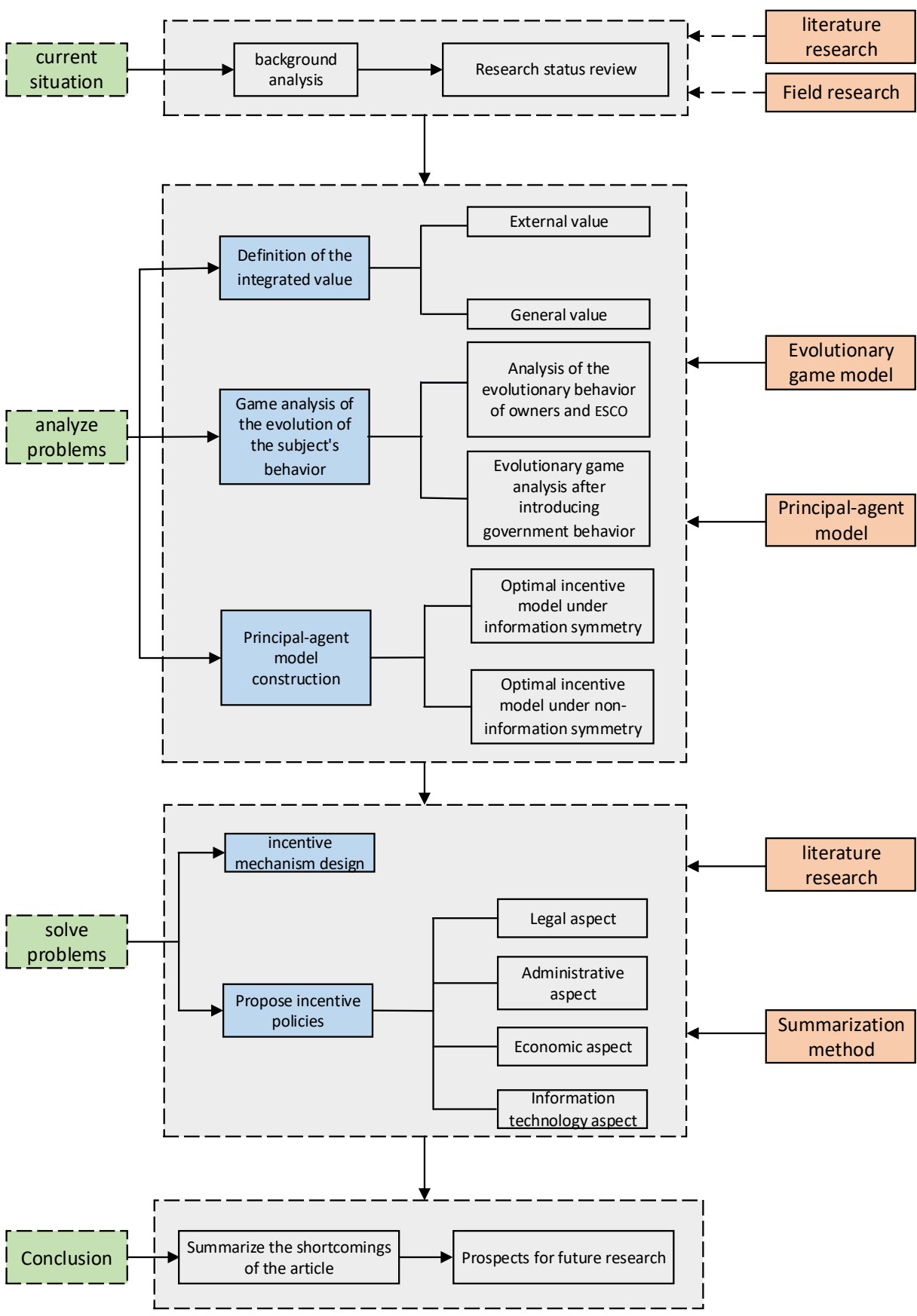

**Figure 1.** Technology roadmap.

*3.2. Application of Evolutionary Game*

There are numerous participants in the monitoring of energy consumption in public buildings. In this research, an evolutionary game model was applied to analyze how the strategic choices of the government, the public building owner or user, and Energy Service Companies (ESCOs) might change following the changes of different variables. Of these, the government is usually the organizer and active enabler of the formulation and direction of energy efficiency in buildings. Research institutes produce systems, analyze surveillance results and the amount of data, and are data users. The public building owner or user is the monitor, who is also the maintainer of the monitoring equipment and is responsible for maintaining stable data uploads. ESCOs are active participants in the building energy efficiency market as third-party players with strong expertise. Thus, governments and research institutions are of the same character in all four cases. As long as the government can formulate policies that fulfill the purpose of continuous monitoring to ensure continuous data collection, research institutions can meet their needs; then, they may not be included in the game. In recent years, evolution game theory (EGT) has been applied with certain concepts, such as the evolutionary stable strategy (ESS) and the replicator equations. EGT is especially appropriate for this study for two reasons, which are as follows: firstly, EGT assumes that players function according to bounded rationality in an environment of incomplete information and that the ESS is not the absolute existence. Secondly, compared to the classical game model, EGT is more advantageous when analyzing the dynamics of a strategy change. Governments, owners, and ESCOs would change their strategy choices over time for responding to the others′ strategy choices. Therefore, EGT fits well with this issue.

To grasp the problems faced by the monitoring of energy consumption in public buildings during their operational phases, we conducted surveys in Nanjing, Yangzhou, Shanghai, and other places in China from August to December 2022. We apply the game analysis model to construct an evolutionary game model of multiple subjects, including owners, ESCOs, and governments. First, we build an evolutionary game model of owners and ESCOs, which are the two key subjects involved in the monitoring of energy consumption in public buildings. Then, we refine and define the cost and benefit parameters that affect the payoff matrix and temporarily analyze its stability strategy and game outcomes. In addition, because the relevant technical management means were not yet mature in the early development of monitoring of energy consumption in public buildings, the benefits of monitoring energy consumption in public buildings and the appropriate reference measures were not recognized by the owners. ESCOs also do not actively choose to provide CMECPB operation and maintenance (O&M) services in the face of extensive uncertainty, thus relying solely on market forces to implement CMECPB, whose data collection is inefficient or even ineffective. Therefore, the government needs to adopt the necessary incentives to motivate owners and ESCOs. Thus, government action is introduced in our work to examine the impact of government incentive policies on the building energy-saving market. Finally, we investigate how and to what extent to incentivize by constructing the principal-agent model. The technical route of research to analyze the excitation mechanism by constructing an evolutionary game model is illustrated in Figure 2.

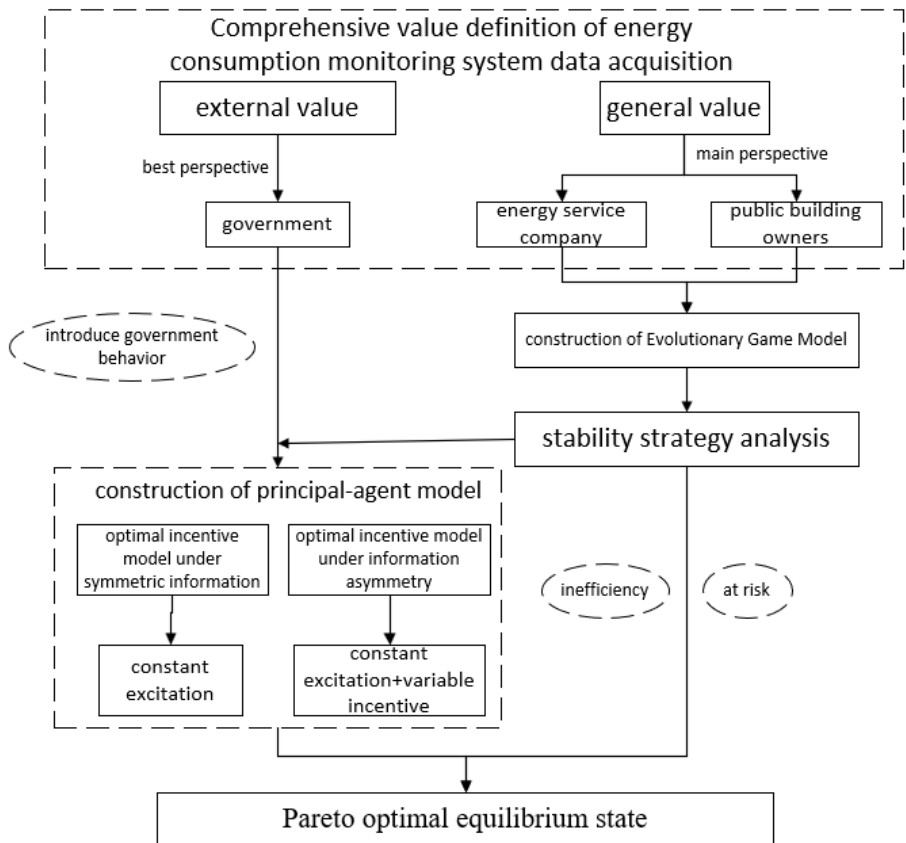

**Figure 2.** The technical route of dynamic game analysis.

### 3.3. Problem Description and Basic Assumptions

3.3.1. Definition of the Integrated Value of Energy Consumption Monitoring System Data Acquisition

The integrated value of CMECPB refers to the real value of the ongoing acquisition of energy consumption monitoring data over the life of the project. The value of goods and items is reflected in their effectiveness and functionality, and the functionality of building energy monitoring is key. Therefore, in this paper, we analyze the functionality of a continuous data acquisition program for the CMECPB project. First, CMECPB projects have the function of reducing energy consumption in buildings, achieving energy conservation in public buildings, obtaining energy-saving benefits, and reducing energy costs. Second, CMECPB projects have public goods that can improve the environment, lead to the development of different industries, and enhance people's quality of life, which can reflect their ecological and environmental functions.

Based on the above considerations, this paper analyzes and defines the composition of the integrated value; the net present value reflects the general value of the project as a function of reducing energy costs and achieving energy savings. Overall, the social and environmental values reflected in the ecological service functions of the project are collectively referred to as the external value of the project. Thus, its comprehensive value includes both general and extrinsic values.

The optimal perspective between the general and external values of the item is shown in Figure 3.

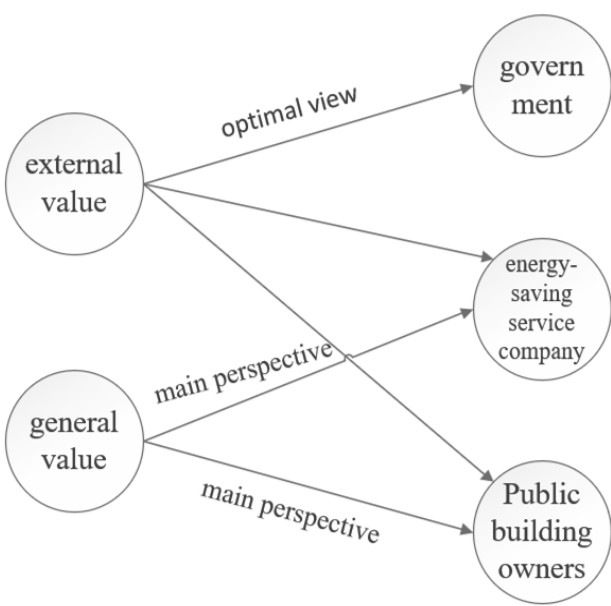

**Figure 3.** Optimal perspectives of each interest body.

### 3.3.2. Basic Ideas and Assumptions

The following assumptions are established for the behavioral game between owners and ESCOs: (i) To achieve equilibrium, owners and ESCOs as a bounded rationality group can adjust their strategic decisions through continuous experimentation, imitation, and learning in the operation and management of public buildings; (ii) Owners and ESCOs have different strategic decision-making spaces. The owners′ strategy space is (support CMECPB, does not support CMECPB), and the ratios of the selected two strategies are x and 1-x; the strategy space of ESCOs is (provide O&M services for CMECPB, no service provided), and the ratios of the two strategies selected are y and 1-y.

Based on the characteristics of the owners and ESCOs, our parameter design for the model is shown in Table 1 below.

**Table 1.** Model parameters and implications.

| Parameter | Meaning |
|:---:|:---:|
| $m_0$ | When ESCOs do not provide O&M services to CMECPB, the benefits to the owner are reduced. |
| $n_0$ | Benefits to ESCOs when the owner does not support CMECPB. |
| $m_1$ | When ESCOs provide O&M services to CMECPB, there is an economic benefit to the owner. |
| $m_2$ | When ESCOs provide O&M services to CMECPB, the owner receives additional revenue for supporting the CMECPB. |
| $n_1$ | When the owner supports CMECPB, the benefits to ESCOs of providing O&M services to CMECPB are increased. |
| $n_2$ | The excess revenue that ESCOs can earn by providing O&M services to a CMECPB is due to the cooperation of the owner when they choose to support the CMECPB. |
| q | Cost for ESCOs to provide O&M services to CMECPB. |
| $q'$ | When supporting CMECPB, there is an additional cost paid by the owner for selecting satisfactory ESCOs. |

*3.4. Model Construction*

3.4.1. Dynamic Game Model Construction between Owners and ESCOS

Based on the definitions of the relevant parameters and assumptions, the game payoff matrix between ESCOS and owner groups can be derived in Table 2 below.

**Table 2.** Game payoff matrix for ESCOS and owners.

| Owners \ ESCOs | Provide CMECPB O&M Services y | No CMECPB O&M Services 1-y |
|---|---|---|
| Support CMECPB x | $m_1 + m_2 - q'$, $n_1 + n_2 - q$ | $m_0 - q'$, $n_0$ |
| Do not support CMECPB 1-x | $m_1$, $n_0 - q$ | $m_0$, $n_0$ |

3.4.2. Introducing Government Actions

The provision of CMECPB project O&M services by ESCO to public building owners requires appropriate management techniques and tools, as it can increase ESCOS' costs and result in a dramatic increase in ESCOS' upfront investment and risk. If the current and future economic benefits are unclear, ESCOS will tend to act negatively and have no power on the supply side. However, due to the constraints of their primary business, owners still lack a certain level of attention to CMECPB and have no means to understand the benefits that CMECPB projects can bring to them. Owners are also gradually becoming more negative, with no incentive on the demand side. Government incentives are an essential means of promoting macro-control. Such economic incentive policies have played an essential role in the development of energy-efficient buildings in various countries. In the United Kingdom, several changed policies have affected the actions and agency of actors, which also affect the transition towards low-energy buildings. Active energy efficiency policies promote the development of intermediaries, and existing organizations have adopted intermediary functions to advance low-energy homes in response to the policies [59]. In California, building energy efficiency was achieved in LAC (Los Angeles County) by means of state policy regulation. The results show the importance of designing energy efficiency policies for growing cities [60]. Moreover, CMECPB is one of the core aspects of the development of energy efficiency in buildings around the world. As one of the leaders and beneficiaries of CMECPB, governments must provide financial incentives to CMECPB participants. Therefore, in this paper, we introduce government behavior regarding economic incentives, both positive and negative, to study the impact of government actions on the development paths of owners and ESCOS.

We assume that a government's positive economic incentive for owners to support CMECPB is $\pi_1$, and the economic penalty for owners not supporting CMECPB is a. The government's positive economic incentive for ESCO to provide O&M services for CMECPB is $\pi_2$, and the economic penalty for ESCO not providing O&M services for CMECPB is b. The game payoff matrix for the owner and ESCO after adding the government financial incentives is shown in Table 3.

**Table 3.** The game payoff matrix between owners and ESCOS with the introduction of government.

| Owners \ ESCOs | Provide CMECPB O&M Services y | Do Not Provide CMECPB O&M Services 1-y |
|---|---|---|
| Support CMECPB x | $m_1 + m_2 - q' + \pi_1$, $n_1 + n_2 - q + \pi_2$ | $m_0 - q' + \pi_1$, $n_0 - b$ |
| Do not support CMECPB 1-x | $m_1 - a$, $n_0 - q + \pi_2$ | $m_0 - a$, $n_0 - b$ |

3.4.3. Principal-Agent Model Construction

In the process of energy consumption monitoring project implementation, there is an obvious information asymmetry between the government and the incentive target, and it is difficult for the government to gain insight into the real implementation cost and

energy-saving effect of the incentive target, so it is the principal. However, ESCOs have more access to gain information in this process and therefore act as the agent.

The government as the client wants to promote the CMECPB project and achieve building energy saving, which is risk-neutral, while ESCOs as the agent are risk-averse in order to maximize their benefits. The CMECPB project can produce comprehensive value V during its run-in phase, and the external value of the project is also directly related to the conventional value of the project. Therefore, the model mainly considers the conventional value of the project, and the different values of the project will be fully considered in the analysis of the model results. Its integrated value is positively related to the agent's effort e, but it is also affected by uncertainties.

$$V = ie + \theta \tag{1}$$

where i denotes the agent's effort to implement the CMECPB, e denotes the conversion coefficient to the integrated value, and the uncertainty $\theta$ of the internal and external environment of the project is represented by the mean of 0 and variance $\sigma^2$. Thus, the integrated value created by the agent depends only on the degree of effort e and is independent of external uncertainty $\theta$, and the variance of the comprehensive value output is $\sigma^2$.

According to the incentive compatibility constraint theory, a linear relationship is assumed between the principal's incentive to the agent, and the comprehensive value generated by the agent's implementation of the CMECPB project, such as

$$S(V) = k + \beta V \tag{2}$$

where k denotes a certain fixed subsidy given to the agent by the government; $\beta$ denotes the shift subsidy to the agent due to the comprehensive value of each unit. When $\beta = 0$, the agent is not required to bear the risk of CMECPB project implementation; when $\beta = 1$, the agent is required to assume the full risk of the CMECPB project.

We suppose that the government utility function is expressed as follows:

$$P[V - S(V)] \tag{3}$$

However, the government is risk-neutral, so the demanded benefits are equal to the demand:

$$E\{P[V - S(V)]\} = E[V - S(V)] = -k + (1 - \beta) E(V) = -k + (1 - \beta) \tag{4}$$

Suppose that the agent's utility function is risk-averse, i.e., $v = -e^{-\rho\omega}$, where $\rho$ denotes the degree of risk aversion, and $\omega$ denotes the actual capital income received by the agent from the CMECPB. The implementation cost of the agent is c (e) = $\frac{1}{2}be^2$; b is also the cost effort coefficient of the agent when b > 0. The greater the value of b, the greater the agent's cost and the greater the negative effect when the effort e is constant. Then, the actual gain of the agent will be

$$W = S(V) - c(e) = k + \beta V - \frac{1}{2}be^2 = k + \beta(ie + \theta) - \frac{1}{2}be^2 \tag{5}$$

Since the agent is risk-averse, the equivalent payoff of the agent should be based on the original benefit minus the cost of risk $\frac{1}{2}\rho\beta^2\sigma^2$; when $\beta = 0$, the cost of risk is 0.

$$CE = E(w) - \frac{1}{2}\rho\beta^2\sigma^2 = k + i\beta e - \frac{1}{2}be^2 - \frac{1}{2}\rho\beta^2\sigma^2 \tag{6}$$

Let $\varpi$ denote the effectiveness of the agent's work outside the CMECPB project. When $CE < \varpi$, the agent will not receive the incentive and therefore should satisfy the incentive compatibility constraint, i.e.,

$$\text{s.t (IR) } k + i\beta e - \frac{1}{2}be^2 - \frac{1}{2}\rho\beta^2\sigma^2 \geq \varpi \tag{7}$$

1. Optimal incentive model under information symmetry

If the government can obtain the degree of effort e of the agent to perform the CMECPB project, that is, the information of the two sides is symmetrical, the incentive compatibility constraint loses its effect, and the government does not need to give the agent additional preferential policies, so the principal-agent model becomes

$$\text{MaxE}\{P[V - S(V)]\} = -k + (1 - \beta) \tag{8}$$

$$\text{s.t (IR) } k + i\beta e - \frac{1}{2}be^2 - \frac{1}{2}\rho\beta^2\sigma^2 \geq \varpi \tag{9}$$

Incentive constraints (IR) can be transformed into

$$-k \leq i\beta e - \frac{1}{2}b^2\sigma^2 - \frac{1}{2}\rho\beta^2\sigma^2 - \varpi \tag{10}$$

Then, the optimal incentive problem becomes

$$\max_{\beta,\, e}\left( ie - \frac{1}{2}be^2 - \frac{1}{2}\rho\beta^2\sigma^2 - \varpi \right) \tag{11}$$

Given $\varpi$ and the derivatives concerning e and $\beta$, respectively, this yields

$$e* = \frac{i}{b}; \quad \beta* = 0 \tag{12}$$

Bringing this condition into the constraint yields

$$k* = \varpi + \frac{i^2}{2b} \tag{13}$$

The derivation of the optimal incentive pattern is

$$S*(V) = k* + \beta*V = k* = \varpi + \frac{i^2}{2b} = \varpi + \frac{1}{2}b(e*)2 \tag{14}$$

The equation above is the Pareto-optimal incentive contract. Since the information is symmetric, the government can observe the principal's effort to implement energy monitoring in public buildings e; when the government observes that e < e*, then the government's incentive is S < S*; when S < $\varpi$, the principal must increase the effort e until e* = $\frac{i}{b}$. Thus, in the case of information symmetry, the principal takes no risk. The optimal incentive model for an incentive subject government is a fixed incentive in the absence of variable indirect incentives:

$$\text{CMECPB optimal incentive model = fixed incentive} \tag{15}$$

2. Optimal incentive model under non-information symmetry

Currently, CMECPB faces information asymmetry under which the government cannot perfectly judge the magnitude of the e-value and achieve Pareto optimization, so the incentive compatibility condition is valid.

Subsequently, the agent's incentive compatibility constraint is as follows:

$$\max_{k,\beta,e} \left( k + i\beta e - \frac{1}{2}be^2 - \frac{1}{2}\rho\beta^2\sigma^2 \right) \tag{16}$$

The first derivative of e in Equations (3)–(23) gives

$$e* = \frac{i\beta}{b} \tag{17}$$

Thus, the optimal incentive model for the principal government changes to

$$\max_{k,\beta,e} E\{P[V - S(V)]\} = -k + (1 - \beta)ie \tag{18}$$

$$\text{s.t (IR) } k + i\beta e - \frac{1}{2}be^2 - \frac{1}{2}\rho\beta^2\sigma^2 \geq \varpi \tag{19}$$

$$\text{(IC) } e = \frac{i\beta}{b} \tag{20}$$

If Equations (11) and (12) are brought into Equation (10), the optimal problem can be expressed as

$$\max_{\beta} \left( \frac{i^2\beta}{b} - \frac{i^2\beta^2}{2b} - \frac{1}{2}\rho\beta^2\sigma^2 - \varpi \right) \tag{21}$$

Then, we find the first derivative of $\beta$ in Equations (3)–(28) as

$$\beta = \frac{i^2}{i^2 + b\rho\sigma^2} > 0 \tag{22}$$

## 4. Results

### 4.1. Analysis of Game Results between Owners and ESCOs

By analyzing the game between owners and ESCOs, we can find that the strategic choice and payoff matrix of owners and ESCOs are related to the payoff matrix and strategic choice of the other party. According to the evolutionary stable strategy of the owners and ESCOs, it is known that a favorable interaction is formed and a Pareto-optimal equilibrium is reached only when, according to $x > x^*$, $y > y^*$, the owners decide to support the CMECPB and the group of ESCOs decides to provide O&M services to the CMECPB, a favorable interaction occurss and a Pareto-optimal equilibrium is reached. Therefore, in order to decrease the value of $x^*$, $y^*$, we should increase $m_2, n_1, n_2$, decrease $q'$, q, increase $n_1 + n_2 - n_0$, increase the payoff level of the Pareto-optimal equilibrium, and increase $x$, $y$. Therefore, at the current stage of CMECPB development, it is inefficient and risky to change the above parameters only by relying on the owners and ESCOs to reach the Pareto equilibrium. This leads both the owners and ESCOs to view CMECPB negatively.

#### 4.1.1. Analysis of Owners' Stability Strategy

Based on the payoff matrix of the game between ESCOs and owners, the owners' payoff is determined to support CMECPB:

$$U_{11} = y(m_1 + m_2 - p) + (1 - y)(m_0 - q) \tag{23}$$

The payoff for owners who choose not to support the CMECPB for building energy consumption is

$$U_{12} = ym_1 + (1 - y)m_0 \tag{24}$$

The average expected gain among the owners' group is

$$U_1 = xU_{11} + (1-x)U_{12} \tag{25}$$

When the owner chooses to support CMECPB, the replicator dynamics equation will be

$$F(x) = \frac{dx}{dt} = x(U_{11} - U_1) = x(1-x)(m_2 y - q') \tag{26}$$

Let $\frac{dx}{dt} = 0$, which yields $x_1^* = 0$, $x_2^* = 1$, $y^* = \frac{q}{m_2}$.

When $y = y^*$, $\forall x$, $F(x) \equiv 0$, $F'(x) \equiv 0$, this means that when the proportion of ESCOs who choose to provide O&M services for CMWCPB reaches $y^* = \frac{q}{m_2}$, which is the proportion of the owners' group that choose to support CMWCPB, and does not reach equilibrium;

When $y > y^*$, $F'(0) > 0$, $F'(1) < 0$, $x_2^* = 1$ is the evolutionary stable strategy of the owners' group; this means that positive interactions are formed, and Pareto-optimal equilibrium is reached when the ESCOs choose to provide O&M services to the CMWCPB and the owners support CMECPB.

When $y < y^*$, $F'(0) < 0$, $F'(1) > 0$, $x_1^* = 0$ is also an evolutionary stable strategy for the owners' group, which means that the proportion of ESCOs that chooses to provide O&M services to the CMWCPB does not reach the start-up point, and the owners who initially support CMWCPB eventually tend not to support CMWCPB.

### 4.1.2. Analysis of ESCOs' Stability Strategy

For ESCOs, the payoff for their choice to provide O&M services to CMECPB is

$$U_{21} = x(n_1 + n_2 - q) + (1-x)(n_0 - q) \tag{27}$$

The payoff to ESCOs for choosing not to provide energy-consuming CMECPB O&M services is

$$U_{22} = xn_0 + (1-x)n_0 \tag{28}$$

The average expected gain of the ESCOs' group is

$$U_2 = yU_{21} + (1-y)U_{22} \tag{29}$$

when ESCOs choose to "provide CMECPB O&M services", the replicator dynamics equation will be

$$F(y) = \frac{dy}{dt} = y(U_{21} - U_2) = y(1-y)[x(n_1 + n_2 - n_0) - q] \tag{30}$$

Let $\frac{dy}{dt} = 0$; then, we obtain $y_1^* = 0$, $y_2^* = 1$, $x^* = \frac{q}{n_1 + n_2 - n_0}$.

When $x = x^*$, $\forall y$, $F(y) \equiv 0$, $F'(y) \equiv 0$, this means that when the proportion of owners that choose to support CMECPB reaches $x^* = \frac{q}{n_1 + n_2 - n_0}$, the proportion of ESCOs that choose to provide O&M services to the CMECPB and those that choose not to provide O&M services to the CMECPB are balanced.

When $x > x^*$, $F'(0) > 0$, $F'(1) < 0$, $y_2^* = 1$ is an evolutionary stable strategy for the ESCOs population, meaning that the ESCOs' choice to provide O&M services for the CMECPB appropriately interacts with the owners' choice to support CMECPB and to achieve Pareto-optimal equilibrium.

When $x < x^*$, $F'(0) < 0$, $F'(1) > 0$, $y_1^* = 0$ is the evolutionary stable strategy for the ESCOs population; this means that the percentage of owners who support CMECPB does not reach the start-up point, and the EDCOs that start to consciously provide O&M services to the CMECPB eventually tend not to provide these services.

*4.2. Analysis of the Effects of Introducing Government Actions*

According to the logic of behavioral science, human behavior results from the combination of demand and motivation. Demand is influenced by external stimuli, which in turn affects human behavior. Therefore, it is necessary to rely on the "tangible hands" of the government to intervene appropriately and necessarily in the operational phase to maximize the benefits for owners and the ESCO society as a whole. The analysis of the model parameters shows that the government needs to guide the parameter values through management incentive policies, such as technical support, reputation incentives, talent development, publicity, and training, to evolve the system into Pareto-optimal equilibrium.

4.2.1. Analysis of Owners′ Stability Strategy after Introducing Government Actions

According to the payoff matrix of the game in which the government action is introduced, the payoff of owners who choose to support the CMECPB is

$$U_{11}' = y(m_1 + m_2 - q' + \pi_1) + (1 - y)(m_0 - q' + \pi_1) \tag{31}$$

The payoffs of owners who choose not to support the CEMCPB are

$$U_{12}' = y(m_1 - a) + (1 - y)(m_0 - a) \tag{32}$$

The average expected gain of the owners′ group is

$$U_1' = xU_{11}' + (1 - x)U_{12}' \tag{33}$$

The replicator dynamics equation for the case of "supporting CMECPB" for the owners′ group is

$$\text{F}(x) = \frac{dx}{dt} = x(U_{11}' - U_1') = x(1 - x)(m_2y - q' + \pi_1 + a) \tag{34}$$

Let $\frac{dx}{dt} = 0$ be $x_1^* = 0$, $x_2^* = 1$, $y^{**} = \frac{q' - \pi_1 - a}{m_2}$.

When $y = y^{**}$, $\forall x$, $\text{F}(x) \equiv 0$, $\text{F}'(x) \equiv 0$, that is, the equilibrium is reached when the proportion of ESCOs that choose to provide CMECPB O&M services reaches $y^{**} = \frac{q' - \pi_1 - a}{m_2}$, and the proportion of owners that choose to support CMECPB and those who do not support CMECPB is equal.

When $y > y^{**}$, $\text{F}'(0) > 0$, $\text{F}'(1) < 0$, $x_2^* = 1$ is the evolutionarily stable strategy of the owners′ group, that is, the owners′ choice to support CMECPB forms a favorable interaction with the ESCOs′ choice to provide CMECPB O&M services, and Pareto-optimal equilibrium is reached.

When $y < y^{**}$, $\text{F}'(0) < 0$, $\text{F}'(1) > 0$, $x_1^* = 0$ is the evolutionarily stable strategy of the owner group, that is, the fraction of ESCOs that choose to provide CMECPB O&M services do not reach the start-up point, and the owners who initially support CMECPB eventually tend not to support it.

At this point, it becomes clear from $y^{**} = \frac{q' - \pi_1 - a}{m_2} < y^* = \frac{q'}{m_2}$ that the strategic conditions for ESCO to choose to provide CMECPB O&M services under government economic incentives are more easily met, and that the percentage of owners who ultimately choose to support the CMECPB strategy is larger.

4.2.2. Analysis of ESCOs′ Stability Strategy after Introducing Government Actions

For ESCOs, the payoff for their choice to provide CMECPB O&M services is

$$U_{21}' = x(n_1 + n_2 - q + \pi_2) + (1 - x)(n_0 - q + \pi_2) \tag{35}$$

The payoff of ESCOs choosing not to offer CMECPB O&M services is

$$U'_{22} = x(n_0 - b) + (1 - x)(n_0 - b) \tag{36}$$

The average expected gain of the ESCO group is

$$U'_2 = yU'_{21} + (1 - y)U'_{22} \tag{37}$$

The replicator dynamics equation for the ESCO group with the option "Provide CMECPB O&M services" is

$$F(y) = \frac{dy}{dt} = y(U'_{21} - U'_2) = y(1 - y)[x(n_1 + n_2 - n_0) - q + \pi_2 + b] \tag{38}$$

Let $\frac{dy}{dt} = 0$; then, we obtain $y_1^* = 0, y_2^* = 1, x^{**} = \frac{q - \pi_2 - b}{n_1 + n_2 - n_0}$.

The proportion of the population of ESCOs that choose to provide CMECPB O&M services and the proportion of those that choose not to provide CMECPB O&M services is balanced at $x = x^{**}, \forall y, F(y) \equiv 0, F'(y) \equiv 0$, that is, the proportion of the population of owners that choose to support CMECPB reaches $x^{**} = \frac{q - \pi_2 - b}{n_1 + n_2 - n_0}$.

When $x > x^{**} F'(0) > 0, F'(1) < 0, y_2^* = 1$ is the evolutionally stable strategy of the population of ESCOs, that is, the choice of ESCOs to provide CMECPB O&M services appropriately interacts with the choice of owners to support CMECPB, and Pareto-optimal equilibrium is reached.

When $x < x^{**}, F'(0) < 0, F'(1) > 0, y_1^* = 0$ is the evolutionary stabilization strategy of the population of ESCOs, that is, the fraction of owners who support CMECPB does not reach the starting point, and ESCOs that start to offer CMECPB O&M services consciously end up tending not to offer CMECPB O&M services.

At this point, it is evident from $x^{**} = \frac{q - \pi_2 - b}{n_1 + n_2 - n_0} < x^* = \frac{q}{n_1 + n_2 - n_0}$ that under government economic incentives, the conditions under which owners choose to support CMECPB policies are more easily met, and ESCOs eventually tend to choose to provide CMECPB O&M service policies in a greater proportion.

*4.3. Equilibrium Point Stability Analysis*

Based on the above analysis, it can be concluded that after the introduction of government economic incentives, the game model between owners and ESCO groups only has five equilibrium points: (0,0), (0,1), (1,0), (1,1), and (x**, y**). The Jacobian matrix of the system is as follows:

$$\begin{aligned} J &= \begin{bmatrix} \frac{\partial F(x)}{\partial x} & \frac{\partial F(x)}{\partial y} \\ \frac{\partial F(y)}{\partial x} & \frac{\partial F(y)}{\partial y} \end{bmatrix} \\ &= \begin{bmatrix} (1 - 2x)(m_2 y - q' + \pi_1 + a) & x(1 - x)m_2 \\ y(1 - y)(n_1 + n_2 - n_0) & (1 - 2y)[x(n_1 + n_2 - n_0) - q + \pi_2 + b] \end{bmatrix} \end{aligned} \tag{39}$$

The determinant of $J$ is det $J = a_{11}a_{22} - a_{12}a_{21}$; trace is tr$J = a_{11} + a_{22}$. If the replicator dynamic equation at some equilibrium point makes det $J > 0$ and tr$J < 0$, then the system is evolutionary stable at this equilibrium point, which is the evolutionary stable strategy (ESS) of the system, but only one stable strategy reaches Pareto-optimal equilibrium. The specific dynamic process tends toward whichever equilibrium point depends on the positive and negative values of the dynamic differential equation in the corresponding interval and the initial state of the strategy ratio adopted by both sides of the game. On this basis, the local evolutionary stability of the equilibrium point of the system is analyzed as follows:

Case 1: At this time, the income of an owner supporting energy consumption data collection is less than that of an owner not supporting energy consumption data collection, and the income of an ESCO choosing to provide energy consumption data collection and

operation services is less than that of an ESCO not providing energy consumption data collection and operation services.

Case 2: At this time, the income of an owner supporting energy consumption data collection is less than that of an owner not supporting energy consumption data collection, and the income of an ESCO choosing to provide energy consumption data collection and operation services is greater than that of an ESCO not providing energy consumption data collection and operation services.

Case 3: At this time, the income of an owner supporting energy consumption data collection is greater than that of an owner not supporting energy consumption data collection, and the income of an ESCO choosing to provide energy consumption data collection and operation services is less than that of an ESCO not providing energy consumption data collection and operation services.

Case 4: At this time, the income of an owner supporting energy consumption data collection is greater than that of an owner not supporting energy consumption data collection, and the income of an ESCO choosing to provide energy consumption data collection and operation services is greater than that of an owner not providing energy consumption data collection and operation services.

Conclusion: It can be seen from Table 4 that there is only one evolutionary equilibrium point (0,0) in scenarios 1–3. The strategy of the system is that the owner does not support energy consumption data collection, and the ESCO does not provide energy consumption data collection and operation services. In case 4, there are two evolutionary stable equilibrium points, namely (0,0) and (1,1). The corresponding (owner, ESCO) strategies are (do not support energy consumption data collection, do not provide energy consumption data collection operation and maintenance services) and (support energy consumption data collection, provide energy consumption data collection operation and maintenance services). In this case only, the system has the possibility of evolving to a (1,1) equilibrium point, but whether it can evolve to the (1,1) equilibrium point depends on the initial state of the system. Therefore, the change of parameters has a significant effect on the stability of the system.

**Table 4.** Local stability of equilibrium points.

| Equilibrium Point (x,y) | Case 1: $m_1+m_2-q'+\pi_1<m_1-a;\ n_1+n_2-q+\pi_2<n_0-b$ | | | Case 2: $m_1+m_2-q'+\pi_1<m_1-a;\ n_1+n_2-q+\pi_2>n_0-b$ | | | Case 3: $m_1+m_2-q'+\pi_1>m_1-a;\ n_1+n_2-q+\pi_2<n_0-b$ | | | Case 4: $m_1+m_2-q'+\pi_1>m_1-a;\ n_1+n_2-q+\pi_2>n_0-b$ | | |
|---|---|---|---|---|---|---|---|---|---|---|---|---|
| | Symbol of DetJ | Symbol of TrJ | Stability | Symbol of DetJ | Symbol of TrJ | Stability | Symbol of DetJ | Symbol of TrJ | Stability | Symbol of DetJ | Symbol of TrJ | Stability |
| $P_1$ (0,0) | + | - | ESS | + | - | ESS | + | - | ESS | + | - | ESS |
| $P_2$ (1,0) | - | indeterminacy | Saddle Point | + | + | Unstable | - | indeterminacy | Saddle Point | + | + | Unstable |
| $P_3$ (0,1) | - | indeterminacy | Saddle Point | - | indeterminacy | Saddle Point | + | + | Unstable | + | + | Unstable |
| $P_4$ (1,1) | + | + | Unstable | - | indeterminacy | Saddle Point | - | indeterminacy | Saddle Point | + | - | ESS |
| $P_5$ (x**,y**) | | | | | | | | | | - | 0 | Saddle Point |

*4.4. Analysis of Incentive Model Results*

In a market environment with information asymmetry, the government's optimal incentive model for agents is to take a degree of project risk according to their risk preference. The government needs to provide not only fixed but also variable incentives to the agents, that is, the optimal incentive model can be formulated as

Optimal incentive for CMECPB = Fixed Incentive + Risk Sharing Coefficient × Principal Benefit

The analysis is as follows:

1.  $\frac{\partial \beta}{\partial b} = -\frac{pi^2\sigma^2}{(i^2+bp\sigma^2)^2} < 0$, that is, the ratio of the variable government subsidies β received by the agent to the effort cost coefficient b, is negatively correlated. This way, under other conditions that remain the same, the agent can only receive a higher economic subsidy by reducing the unit effort cost. In this case, public building owners will partner with ESCOs to implement continuous energy consumption monitoring and reduce costs.

2.  $\frac{\partial \beta}{\partial p} = -\frac{bi^2\sigma^2}{(i^2+bp\sigma^2)^2} < 0$, that is, the government shift subsidy ratio β received by the agent is negatively related to the risk aversion coefficient p, that is, the larger the risk aversion coefficient of the agent, the smaller the government economic incentive received. The risk aversion coefficient p will be different for ESCOs and public building owners, and the economic incentives also will be different.

3.  $\frac{\partial \beta}{\partial \sigma^2} = -\frac{bi^2 p}{(i^2+bp\sigma^2)^2} < 0$, that is, the ratio of the variable government subsidies β received by the agent is negatively correlated with the external uncertainty variance $\sigma^2$, indicating that the larger the external uncertainty, the greater the economic incentive received. For public building owners and ESCOs, the condition that external uncertainty reduces their returns is imposed.

4.  $\frac{\partial \beta}{\partial \iota} = -\frac{2b\iota(p\sigma^2)^2}{(\iota^2+bp\sigma^2)^2} > 0$, that is, the ratio of government shift subsidies received by agents β is positively correlated with the composite value transformation factor i. The greater the value created by the agent, the higher the economic incentive it receives. Since there is virtually no cost to the owner in the CMECPB project, the economic incentive to the owner is mainly based on the external value it creates, while the incentive to the section ESCO should be carried out by considering the external value created by it and the regular value of the project.

Therefore, an effective incentive mechanism can improve the effort level of agents and considerably promote the development of CMECPB in various countries. By analyzing the optimal incentive model, the incentive mechanism should be designed while taking into account the agent's cost-benefit coefficient b, risk avoidance ρ, external uncertainty $\sigma^2$, and integrated value transformation coefficient I, among other factors.

## 5. Discussion

*5.1. Incentive Mechanism Design*

In the previous section, the stabilization strategies of different agents for building en-ergy monitoring have been analyzed based on a game model, and the incentive model with the introduction of government actions has been focused on. Theoretically, incentive mechanisms are laws of structure, manner, relationship, and the evolution of interactions and mutual constraints with incentive objects in organizational systems, which utilize multiple means of incentives and make them standardized and relatively fixed to transform ambitious ideals into concrete facts. In this paper, the incentive mechanism is based on the two essential subjects of public building owners and ESCOs. Through the operation and promotion of the whole mechanism, on the one hand, it provides incentive policies as well as policy preferences and support to CMECPB in different aspects. On the other hand, it combines administrative means with vertical, mandatory, and quick-acting characteristics to encourage and stimulate the development of the building energy market. Based on the

above findings, the incentive mechanism is designed in three stages. First, the supervision of government office buildings, schools, and hospitals supported by financial funds is considered. The economic incentives and management mechanisms for commercial construction owners and ESCOs are then proposed to form a sound long-term mechanism for the energy consumption monitoring market. As two finite rational groups, owners and ESCOs can reach an equilibrium state of interest and then achieve Pareto optimality regularly by adjusting their behavior, strategies, etc. Finally, this benign circular state is transformed into endogenous power, which is used to stimulate the motivation and creativity of market entities to monitor energy consumption and achieve goals.

When designing the CMECPB incentive mechanism, different incentive mechanisms should be formulated based on different incentive objects, their value requirements, and the influence factors of optimal incentives.

1.   Incentives for public building owners

In the CMECPB implementation, the cost of implementation is 0, the risk avoidance coefficient ρ is low, the effort cost factor b is close to 0, and some energy-saving benefits can also be achieved. The main reason for the lack of enthusiasm among property owners to participate in CMECPB is their feeble awareness of energy conservation and the psychological role of seeking fairness. Therefore, the incentive model for public building owners should be based on mandatory indicators and mental incentives, such as establishing energy consumption quota indicators and ranking announcement, and supplemented by minor financial subsidies that internalize external values.

2.   Incentives for ESCOs

ESCOs are the implementers of the CMECPB that primarily pursue the conventional value of the project. According to the principal-agent model, the lack of incentive for ESCOs to perform CMECPB is mainly due to the elevated external uncertainty coefficient $\sigma^2$. The cost of the project implementation is steep, but the comprehensive value conversion coefficient i is not obvious. Therefore, ESCOs can be motivated from two aspects: on the one hand, it is necessary to improve its energy-saving benefits through financial subsidies and other economic incentives to achieve external value internalization and NPV > 0. On the other hand, third-party financial institutions can be introduced to improve the energy consumption monitoring market industry chain and reduce the cost coefficient b. The integrated value conversion coefficient I of ESCOs also needs to be improved.

The design of incentive mechanism should be executed in three stages depending on the incentive goal, principle, optimal incentive model, and incentive strength.

The first phase is mainly aimed at encouraging government offices supported by financial funds, and public buildings, such as schools and hospitals that are partly supported by financial funds. Because such buildings are state property or have public welfare attributes, they do not generate economic benefits, and their day-to-day operations depend primarily on state financial allocations. Therefore, the owners′ risk aversion ρ and external variables $\sigma^2$ are negligible, and the effort cost coefficient b is nearly 0. Thus, a range of restrictive measures, such as energy consumption regulation, energy quota, energy consumption promotion, accountability of leadership, and incentive policies, such as spiritual incentives and departmental leadership performance appraisal, can be applied to enable such public buildings to be proactively monitored on an ongoing basis. In addition, for certain commercial building owners who are willing to conserve energy, an energy disclosure system could also have the effect of providing a horizontal comparison of energy consumption.

In the second phase, thanks to the demonstration effect of energy efficiency achievements in government office buildings and the incentive of a series of national and local favorable policies, a growing number of commercial construction owners and ESCOs have joined the energy-saving effort in construction. The most essential task at this time is to strengthen the management of energy efficiency in the main subjects of building energy conservation, namely, the majority of commercial and public buildings. At this stage, in addition to carrying on strengthening the demonstration and the leading role of energy

consumption management in government office buildings, a series of preferential policies should be adopted to encourage the development and growth of ESCOs and energy-saving product suppliers. In this way, high-quality and low-cost energy-saving products and services can be provided for commercial building owners, and their energy-saving cost coefficient b can be reduced to increase their energy-saving benefits. The implementation of such a range of positive incentives, including tax and fiscal incentives, loan discounts, and energy-saving incentives, reduces energy-saving costs b and the uncertainty $\sigma^2$ of exogenous variables. Then, a series of reverse incentive policies needs to be formulated, such as the implementation of a ladder energy consumption price. At this time, this work focuses on regulating the energy-saving market for buildings and the strict monitoring of the mechanism of energy consumption.

In the third phase, markets for monitoring energy consumption should be cultivated and long-term mechanisms created. In this phase, there should be targeted incentives for different incentive targets. The rapid development of service and investment institutions such as ESCOs and suppliers of energy consumption monitoring equipment should be actively encouraged to realize the specialization, science and technology, industrialization, regularization, and high efficiency of the entire energy consumption monitoring market, thus reducing implementation costs and external uncertainties $\sigma^2$. The integrated value transformation coefficient I of the building energy consumption monitoring market will be considerably increased, and the integrated value will be thoroughly explored when the capabilities of the entire market players have been significantly enhanced. In addition to positive incentives, a dedicated, independent, third-party energy audit department should certainly be established to conduct the dynamic audit oversight of energy consumption in public buildings and to analyze the causes of elevated energy consumption promptly. The causes of non-conforming public buildings should be determined to find the best solutions. With the effective cooperation of positive incentives and energy consumption supervision, the market for energy consumption monitoring in public buildings can be effectively promoted into a virtuous circle. Figure 4 shows the incentive mechanism and subsidy design of CMECPB.

### 5.2. Policy Implications

Based on practical studies to classify CMECPB problems and to design evolutionary game analysis and incentive mechanisms, this section analyzes and explores the incentive mechanism of CMECPB in four aspects: policy measures, management measures, economic measures, and technical measures. The basic framework of the proposed incentive policy for CMECPB is illustrated in Figure 5.

CMECPB is essential to obtain stable, continuous access to energy consumption data for public buildings and to develop energy conservation strategies. As tripartite stakeholders, the government, ESCOs, and building owners or users have a significant influence over the CMECPB. Therefore, based on the above findings, the following policy incentive implications are proposed in this paper:

1. In regard to legal policies, normative documents such as CMECPB technical rules and energy consumption standards for public buildings should be improved, and the dynamic management of standard rules should be strengthened. Monitoring data with higher consistency are currently lacking. By setting and closely following operating standards for monitoring systems, the foundation for energy conservation in buildings can be laid.

2. From the aspect of administrative management, it is necessary to clarify the supervision and assessment system, establish a continuous monitoring evaluation mechanism, link the performance of the person in charge, and explore the organizational structure of CMECPB supervision. Large government public buildings, a major energy consumer, account for 5% of China's total annual electricity consumption. The government should play an active and exemplary role in promoting energy-saving management and target management in government office buildings. By utilizing

the reputation incentive, that is, recognizing good owners or users in CMECPB, good ESCOs, and other internal drivers, the propaganda effect of energy conservation in buildings can be formed throughout society.

3.  From the aspect of economic incentives, subsidies and various incentives are provided to CMECPB project suppliers, demanders, and investors. To broaden the scope of ESCOs, the government should set up a special guarantee fund for CMECPB and encourage banks to innovate financial products to reduce their credit difficulties. It is necessary to introduce loan subsidies for owners or users of commercial public buildings to increase the enthusiasm of project owners. In addition, the owners or users and ESCOs should be promptly penalized for their negative behavior.

4.  From the technical support side, a public information platform based on the building energy consumption monitoring industry chain should be actively built to realize resource sharing and information symmetry within the industry chain. The government should combine the practical advantages of ESCOs with the scientific research advantages of universities and research institutes, to improve the management means and technical methods of CMECPB O&M and to reduce the operating costs of CMECPB. At the same time, the "industry-university-research-application" mechanism with the joint participation of ESCOs, scientific research institutions, and universities should be cultivated so that a multi-win-win can be formed by the township.

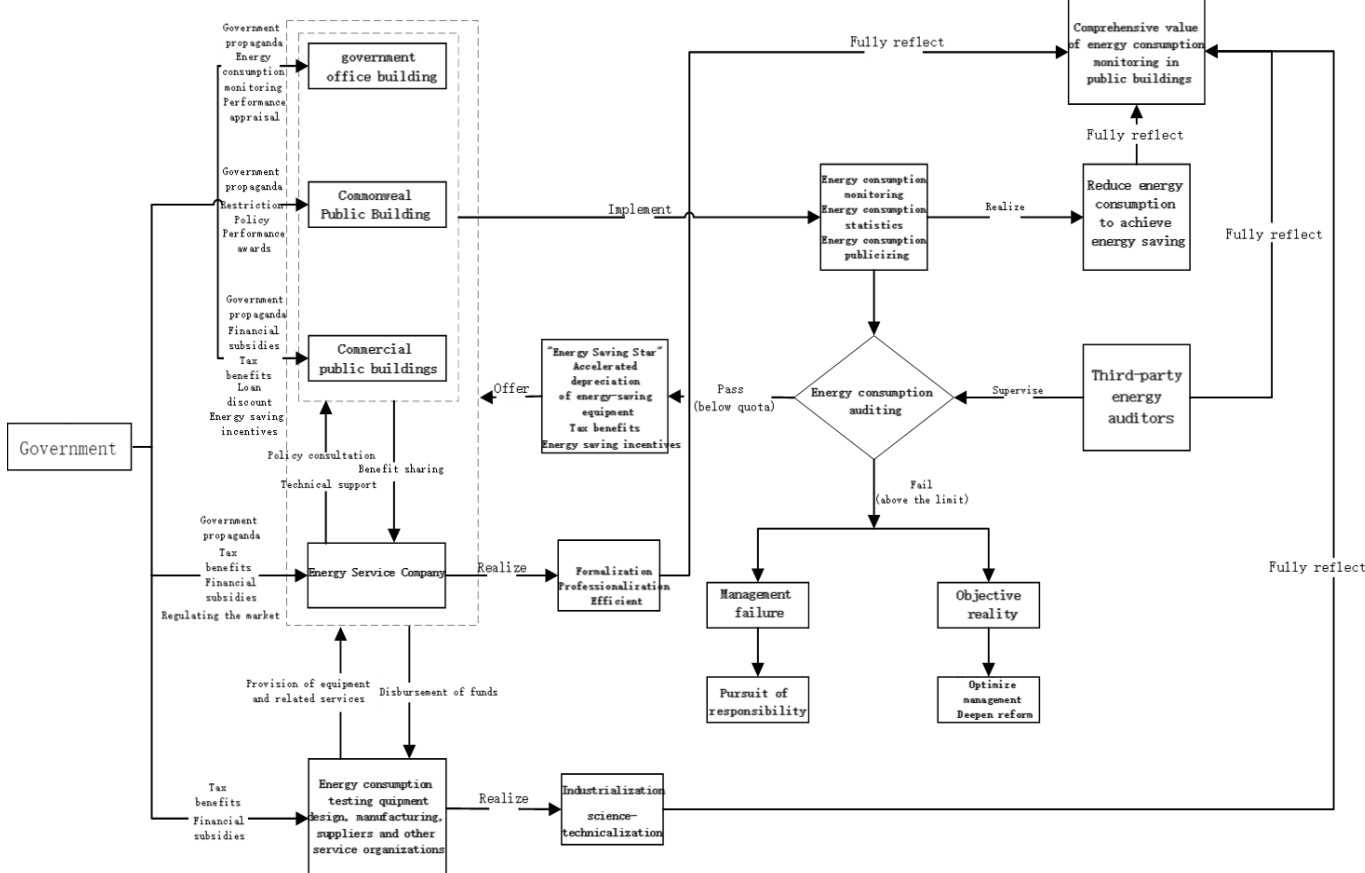

**Figure 4.** CMECPB incentive mechanism and subsidy design.

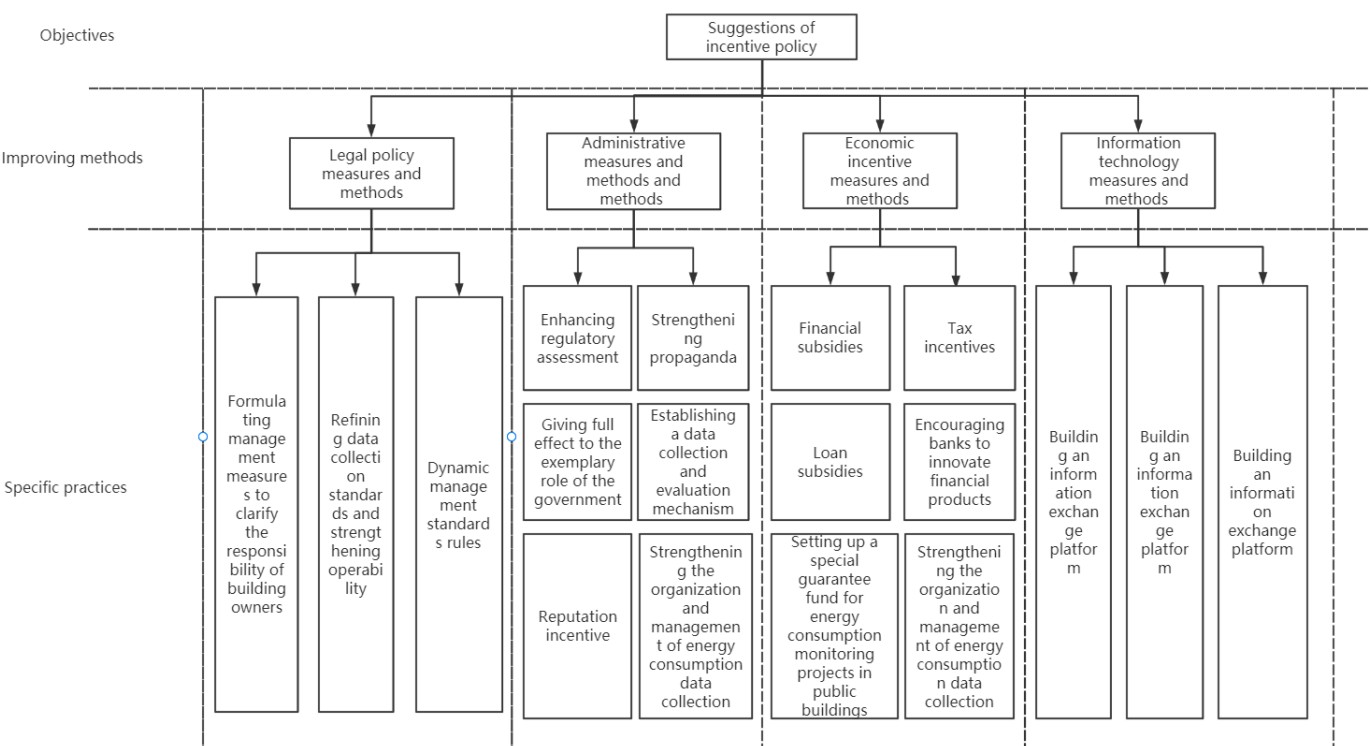

**Figure 5.** The basic framework of CMECPB's incentive policy recommendations.

## 6. Conclusions

The research and discussion of incentive policies and stakeholders' behavioral decisions in the building energy sector not only play a key role in promoting the development of building energy consumption monitoring but also have a critical significance for the optimization of building energy use structure and sustainable transformation of the building industry. In this paper, we innovatively considered ESCOs as one of the subjects of the game and constructed a dynamic game model for the owner, ESCOs and government in the presence of information asymmetry using game analysis models.

The results showed that during the current phase of the operation of CMECPB, the economic incentives of the government and the management incentives should be implemented simultaneously. To achieve equilibrium in the public building energy market, the government can encourage the public building owners or users to engage with the ESCOs simultaneously so that the system can reach the Pareto-optimal equilibrium state as soon as possible. This result is corroborated by related papers from other countries, such as the impact of policy changes on building energy efficiency in the UK and California regions mentioned on page 11. This paper further examined the questions of "who should be motivated" and "how to motivate". The optimal incentive model was constructed using principal-agent theory, and the results showed that the optimal incentives are a "fixed incentive" in the case of information symmetry and a "fixed incentive + variable incentive" in the case of information asymmetry. It was also found that the optimal incentive model is influenced by the cost effort coefficient b, risk aversion $\rho$, external uncertainty $\sigma^2$, and comprehensive value conversion coefficient i at the same time. Therefore, the determination of incentive strength should be based on the general and external values of the project while taking into account multiple factors to be determined. We presented excitation methods for different objects according to their excitation influence factors. The comprehensive design of the economic subsidy was achieved by following the incentive targets. The incentive and subsidy design strategies were proposed according to the incentive mechanism.

Our findings help governments, building owners or users, and ESCOs to better understand the problems and challenges faced in CMECPB, and provide some references and recommendations on how to stabilize and continuously obtain monitoring data on building

energy consumption. Thus, we lay down the basis for the government to develop incentive policies to better promote continuous monitoring of building energy consumption and building energy efficiency. However, this study still has the following limitations and therefore needs additional development: (1) We only studied the field of public buildings and did not involve civil buildings such as residential buildings, nor monitoring equipment suppliers and other stakeholders. The main reason is that public buildings often have a single owner, which is more in line with the traditional game model. Although residential buildings account for a large proportion in the construction field, they often involve multiple owners, and each owner has different strategies. Therefore, the game is difficult and does not apply to the traditional evolutionary game analysis model. At present, there are few studies on energy consumption monitoring of residential buildings, and the research samples are insufficient and lack scientific demonstration. However, as the concept of green building is recognized and accepted by more and more parties, the topic of residential building is a direction worth studying. Future research should combine the process of industry development and technological innovation and consider the diversification of research objects to enrich the research scope. As the concept of green building is recognized and accepted by a growing number of parties, future research should include residential owners who require low-energy buildings. In addition, from a market supply perspective, stakeholders such as suppliers of building energy monitoring equipment can also be considered for a more comprehensive and in-depth discussion. (2) In the development and change of the building energy monitoring market, the government should also adjust the incentive policy according to the actual situation. The complexity of adopting different combinations of strategies, or even making policy updates, has not been considered and is subject to additional investigation and analysis. (3) As China is still in the development stage of building energy monitoring, we focused on the role and responsibility of the government at this stage. However, the market is likely to play an increasingly crucial role as building energy consumption monitoring continues to evolve. Then, the shift in dominance between the government and the market will be a credible future research direction.

**Author Contributions:** Conceptualization, H.C. and G.F.; methodology, H.C.; software, Y.X.; validation, H.C. and Q.L.; formal analysis, H.C. and Y.X.; investigation, Y.X. and G.F.; writing—original draft preparation, H.C. and Q.L.; writing—review and editing, H.C. and G.F.; supervision, G.F. All authors have read and agreed to the published version of the manuscript.

**Funding:** This research is financially supported by Nanjing Green Building and Green Building Materials Development Center.

**Institutional Review Board Statement:** Not applicable.

**Informed Consent Statement:** Informed consent was obtained from all subjects involved in the study.

**Data Availability Statement:** Data collected from the questionnaire survey and the data analysis results presented in the paper are available from the corresponding author by request.

**Conflicts of Interest:** The authors declare no conflict of interest.

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
