# Peer review of "Incentive Mechanism and Subsidy Design for Continuous Monitoring of Energy Consumption in Public Buildings (CMECPB): An Overview Based on Evolutionary Game Theory"

_buildings, doi:10.3390/buildings13040984_

Round 1

Reviewer 1 Report

Interesting paper. Not my technical area but one that deserves investigation (I could see this being applied in my work). 

Just a couple of minor points:

GET is used as the acronym for evolutionary game theory in two places on page 6. Its EGT elsewhere.

The statement on page 8 (line 336/337) about incentive policies playing a role in countries should have a reference (I'm wondering which countries?)

Author Response

    We would like to thank you for your careful reading, helpful comments, and constructive suggestions, which has significantly improved the presentation of our manuscript. We have carefully considered all comments from the reviewers and revised our manuscript accordingly. The manuscript has also been double-checked, and the typos and grammar errors we found have been corrected. In the following section, our responses are given in red font and changes/additions to the manuscript are given in blue text. We believe that our responses have well addressed all concerns from the reviewers. We hope our revised manuscript can be accepted for publication.

Point 1: GET is used as the acronym for evolutionary game theory in two places on page 6. Its EGT elsewhere.

Response 1: We were really sorry for our careless mistakes. Thank you for your reminder. As suggested by the reviewer, we have corrected the “GET” into “EGT” on page 8.

Point 2: The statement on page 8 (line 336/337) about incentive policies playing a role in countries should have a reference (I'm wondering which countries?)

Response 2: Thank you for your excellent suggestion. We have re-written this part according to the Reviewer’s suggestion. On page 11 (line 408-415), we have added the application of incentive policies in specific countries. The specific modifications are as follows:

In the United Kingdom, several changed policies have affected the actions and agency of actors, which also affect the transition towards low-energy buildings. Active energy efficiency policies promote the development of intermediaries, and existing organizations have adopted intermediary functions to advance low-energy homes in response to policy [60]. In the California region, building energy efficiency was achieved in the LAC (Los Angeles, California) through the means of state policy regulation. The results show the importance of designing energy efficiency policies for growing cities [61].

At the same time, we also added relevant expressions in the Conclusion of Part 6, Page 23 (lines 934/935) to verify the scientific nature of our research results, specifically as follows:

This result is corroborated by related papers from other countries, such as the impact of policy changes on building energy efficiency in the UK and California regions mentioned on page 11.

    We tried our best to improve the manuscript and made some changes marked up using the “Track Changes” function in revised paper which will not influence the content and framework of the paper. We appreciate for Editor/ Reviews’ warm work earnestly, and hope the correction will meet with approval. Once again, thank you very much for you comments and suggestions.

Reviewer 2 Report

Results of the findings will be helpful for governments policy measures for lowering building energy consumption. It is necessary to continue your research for other types of buildings, mainly residential buildings.

Author Response

    On behalf of all the contributing authors, I would like to express our sincere appreciations of your letter and reviews’ constructive comments concerning our article entitled “Incentive mechanism and subsidy design for Continuous Monitoring of Energy Consumption in Public Buildings (CMECPB): An overview based on evolutionary game theory”. These comments are all valuable and helpful for improving our article. We have carefully considered your comments and revised our manuscript accordingly. The manuscript has also been double-checked, and the typos and grammar errors we found have been corrected. In the following section, our response is given in red font and changes/additions to the manuscript are given in the blue text. We believe that our responses have well addressed all concerns from the reviewers. We hope our revised manuscript can be accepted for publication.

Point 1: Results of the findings will be helpful for governments policy measures for lowering building energy consumption. It is necessary to continue your research for other types of buildings, mainly residential buildings.

Response 1: We think this is an excellent suggestion. We have added this section on page 23-24 of the Conclusion (lines 957-967) in accordance with your suggestion. In view of this problem, the research limitations in the conclusion part of this paper are as follows :

We only studied the field of public buildings, and did not involve civil buildings such as residential buildings, nor monitoring equipment suppliers and other stakeholders. The main reason is that public buildings are often single-owner, which is more in line with the traditional game model. Although residential buildings account for a large proportion in the construction field, they often involve multiple owners, and each owner has different strategies. Therefore, the game is difficult and does not apply to the traditional evolutionary game analysis model. At present, there are few studies on energy consumption monitoring of residential buildings, and the research samples are insufficient and lack of scientific demonstration. However, as the concept of green building is recognized and accepted by more and more parties, the topic of residential building is a direction worth studying. Future research should combine the process of industry development and technological innovation, and consider the diversification of research objects, so as to enrich the research scope.In addition, from a market supply perspective, stakeholders such as suppliers of building energy monitoring equipment can also be considered for a more comprehensive and in-depth discussion.

    We tried our best to improve the manuscript and made some changes marked up using the “Track Changes” function in revised paper which will not influence the content and framework of the paper. We appreciate for Editor/ Reviews’ warm work earnestly, and hope the correction will meet with approval. Once again, thank you very much for you comments and suggestions.

Reviewer 3 Report

This paper constructs an evolutionary game model for continuous monitoring of energy consumption in public buildings between ESCO and the owner. It studied the dynamic evolution path of a game system and the evolutionary stable strategy under market-based mechanisms. In my opinion, this paper is interesting, but there are some minor comments that must be covered.

There are many similar papers in the literature that investigated the financial incentive exponentially in the supply chain (http://dx.doi.org/10.7232/iems.2014.13.4.383). Authors should compare the innovation of their work with this article in one or two sentences.

I found the use of abbreviations excessive. Please check the manuscript to see if you can get rid of a few. Alternatively, please provide a list of abbreviations in an appendix. The reader could then print this list and consult it when necessary when reading the paper.

The sections of this paper are not conventional for a scientific paper. Particularly the "Results" section lacks.

The conclusion is clear, but should be sustained by a stronger scientific content.

Author Response

     We would like to thank you for your careful reading, helpful comments, and constructive suggestions, which has significantly improved the presentation of our manuscript. These comments are all valuable and helpful for improving our article. As you are concerned, there are several problems that need to be addressed. According to your nice suggestions, we have made extensive corrections to our previous draft. The manuscript has also been double-checked, and the typos and grammar errors we found have been corrected. In the following section, our response is given in red font and changes/additions to the manuscript are marked up using the “Track Changes” function. We believe that our responses have well addressed all concerns from the reviewers. We hope our revised manuscript can be accepted for publication.

Point 1: Dose the introduction provide sufficient background and include all relevant references?(Must be improved)

Response 1: We gratefully appreciate for your valuable comment. According with your advice, we amended the “introduction” part in manuscript. Our original manuscript was limited to the current situation in China, so we researched the global situation and fleshed out the background in order to be more convincing. The specific modifications are as follows:

According to Global ABC's 2022 GLOBAL STATUS REPORT FOR BUILDINGS AND CONSTRUCTION, building energy demand increased by approximately 4% over 2020 to 135 EJ, the largest increase in the last 10 years. This means that CO2 emissions from building operations are at an all-time high of about 10 billion tons of CO2, about 5% more than in 2020 and 2% higher than the previous peak in 2019. Statistically, large public buildings consume high energy, and the area of large public buildings accounts for less than 4% of the building area in China, but the energy consumption accounts for more than 20% of the building energy consumption. International Energy Agency (IEA) released the World Energy Outlook 2022, which reported that the energy supply crisis has had a negative impact on global economic activity. For countries that rely on energy imports, higher energy prices have raised production costs for businesses, leading to lower economic output, while some countries engaged in energy exports have benefited from higher prices.

Point 2: Are all the cited references relevant to the research? (Can be improved)

Response 2: We sincerely appreciate the valuable comments. We have checked the literature carefully and replaced and added some of the literature in the literature review section of the revised draft to support our research. The specific modifications to the literature are as follows:

Substitution of references:

[12] Zhao, Y.; Zhang, C.; Zhang, Y.; Wang, Z.; Li, J. A review of data mining technologies in building energy systems: Load prediction, pattern identification, fault detection and diagnosis. Energy and Built Environment, 2020, 1(2), 149-164. → Yun, G. Y.; Kim, H.; Kim, J. T. Effects of occupancy and lighting use patterns on lighting energy consumption. Energy and Buildings, 2012, 46, 152-158.

[14] Zhu, J.; Li, D. Current situation of energy consumption and energy saving analysis of large public building. Procedia En-gineering, 2015, 121, 1208-1214. → Marinakis, V.; Karakosta, C.; Doukas, H.; Androulaki, S.; Psarras, J. A building automation and control tool for remote and real time monitoring of energy consumption. Sustainable Cities and Society, 2013, 6, 11-15.

[18] Gabbar, H. A.; Musharavati, F.; Pokharel, S. System approach for building energy conservation. Energy Procedia, 2014, 62, 666-675. → Bøhm, B.; Danig, P. O. Monitoring the energy consumption in a district heated apartment building in Copenhagen, with specific interest in the thermodynamic performance. Energy and buildings, 2004, 36(3), 229-236.

Delete [45], change [46] to [45], add new [46]Melbinger, A.; Cremer, J.; Frey, E. Evolutionary game theory in growing populations. Physical review letters, 2010, 105(17), 178101.

[54] Qian, Q. K.; Chan, E. H.; Choy, L. H. How transaction costs affect real estate developers entering into the building en-ergy efficiency (BEE) market? Habitat International, 2013, 37, 138-147. → Yang, X.; Zhang, J.; Shen, G. Q.; Yan, Y. Incentives for green retrofits: An evolutionary game analysis on Pub-lic-Private-Partnership reconstruction of buildings. Journal of cleaner production, 2019, 232, 1076-1092.

Change here on lines 232-237 of the original:

Yang et al. [54] reveals the game strategy change of encouraging green retrofits and implementing green retrofits in government groups and investment groups through an evolutionary game analysis. Research shows that the increased benefits and reduced costs have the best incentive effectiveness. The positive policy incentive measure will reduce the amount of investment groups which implement green retrofits, while the negative policy incentive measure will achieve a high level of effectiveness.

Supplement to the references (here added on lines 250-254 of the original manuscript) :

Zhou et al. [57] used evolutionary game theory to build green technology innovation activities of the government, public, polluting enterprises, and non-polluting enterprises of a four-group evolutionary game model under environmental regulations to discuss and analyze the strategic stability of each game subject and the influence mechanism of strategic selections.

[57] Zhou, X.; Jia, M.; Wang, L.; Sharma, G. D.; Zhao, X.; Ma, X. Modelling and simulation of a four-group evolutionary game model for green innovation stakeholders: Contextual evidence in lens of sustainable development. Renewable Energy, 2022, 197, 500-517.

Point 3: Is the research design appropriate?(Must be improved)

Response 3: We think this is an excellent suggestion. In response to this problem, we have improved some of the contents of the article research, and added a research design roadmap in 3.1 to make the research design of this article more clear. The specific modifications are as follows:

3.1. Research design and model usability analysis

The whole research framework of this paper is clear. Based on the full analysis of the background and the summary of the research status, the qualitative and quantita-tive analysis is carried out for the difficulty of collecting energy consump-tion moni-toring data. The evolutionary game model and the principal-agent model are used to analyze the problem of whether to motivate and how much to motivate. Final-ly, a so-lution to the difficulty of collecting energy consumption monitoring data is proposed, an incentive mechanism is designed, and corresponding incentive policy recommen-dations are given. The technical roadmap of this paper is shown in Figure 1.

Figure 1. technology roadmap

The incentives for CMECPB are designed to address the lack of incentives for con-tinuous monitoring and to promote the marketization of building energy efficien-cy, which will alleviate and reduce the problem of inconsistent data used for building energy consumption decision-making. Building energy-saving also has strong positive externalities, that is, building energy conservation can bring additional social and en-vironmental benefits. Therefore, energy conservation and environmental protection will be the main themes in the future development of the building energy conservation market, which in turn will promote the continuous renewal and improvement of in-centive means. Evolutionary game models based on game analysis provide a frame-work for elaborating incentive policies for different disciplines by finding the best perspective on the general and external value of a project to the government, the own-ers, and the ESCOs. These models appear to be valuable in fully demonstrating where the main potential for technology-related continuous monitoring lies in the relevant sector of continuous monitoring of building energy consumption, which is useful for identifying practical constraints that can be expected from policy measures.

Point 4: There are many similar papers in the literature that investigated the financial incentive exponentially in the supply chain (http://dx.doi.org/10.7232/iems.2014.13.4.383). Authors should compare the innovation of their work with this article in one or two sentences.

Response 4: Thank you for your excellent suggestion. According to your comments, we have carefully studied the differences between the design of incentive mechanism in supply chain and that in this paper, and learned related content. To solve this problem, we have made a supplement in lines 255-274 on page 5-6 of the literature review, which is as follows:

In addition, evolutionary game theory is also widely used in logistics network system, supply chain and so on. Mohammad Asghari et al. [58] investigated a dynamic mathematical model to evaluate the incentive effect on return quantity of used products in a reverse logistics network design, which included different stages of managing waste product distribution through coordination between collection centers and recovery facilities to warrant economic efficiency. The innovation lies in a fuzzy approach applied in the optimization model to interpret the relationship between the rate of return and the suggested incentives. Wan et al. [59] established a cooperative poverty alleviation model with multi-subject participation composed of cooperative, smart supply chain platform and the government. The evolutionary game method was used to explore the cooperative poverty alleviation strategy of smart supply chain platforms and cooperatives under the government financial platform subsidy mechanism. Particularly, the game equilibrium of the smart supply chain platform and cooperative in cooperation with and without the government financial platform subsidy was analyzed. This paper applies game theory to the monitoring of energy consumption in public buildings. The innovation is that this paper considers the incentive target, value re-quirement and the influencing factors of the optimal incentive according to the results of the evolutionary game of stakeholders; and by constructing an incentive model, the incentive mechanism is designed in a more targeted way, which provides institutional guarantee for the government side to promote the energy consumption monitoring of public buildings.

Point 5: I found the use of abbreviations excessive. Please check the manuscript to see if you can get rid of a few. Alternatively, please provide a list of abbreviations in an appendix. The reader could then print this list and consult it when necessary when reading the paper.

Response 5: We sincerely appreciate the valuable comments. We examined the manuscript carefully, and there were some abbreviations. As you suggested, we have removed some unnecessary abbreviations, and only 5 abbreviations of commonly used words in the article have been retained in the revised manuscript, which are: Continuous Monitoring of Energy Consumption in Public Buildings (CMECPB), Energy Service Companies (ESCOs), Building Energy Management Systems (BEMS), Evolution Game Theory (EGT), and Operation and Maintenance (O&M).

Point 6: The sections of this paper are not conventional for a scientific paper. Particularly the "Results" section lacks.

 Response 6: In response to this problem, we found that the conclusion of the evolutionary game model lacks the stability analysis of the equilibrium point by learning similar papers and methods, so we added the content of the equilibrium point analysis in 4.3. Thank you very much for the valuable comments of the reviewers, let us further improve this paper. The specific modifications are as follows:

4.3. Equilibrium point stability analysis

Based on the above analysis, it can be concluded that after the introduction of government economic incentives, the game model between owners and ESCO groups has and only has five equilibrium points: (0,0)、(0,1)、(1,0)、(1,1)、(x**, y**). The Jacobian matrix of the system is as follows:

If the replicator dynamic equation at some equilibrium point makes det J>0 and trJ<0, then the system is evolutionary stable at this equilibrium point, which is the evolutionary stable strategy ( ESS ) of the system, but only one stable strategy is Pareto optimal equilibrium. The specific dynamic process tends to which equilibrium point depends on the positive and negative values of the dynamic differential equation in the corresponding interval and the initial state of the strategy ratio adopted by both sides of the game. On this basis, the local evolutionary stability of the equilibrium point of the system is analyzed as follows:

Table 4. Local stability of equilibrium points.

Case 1

Case 2

Case 3

Case 4

Equilibrium point

(x,y)

Symbol of DetJ

Symbol of TrJ

stability

Symbol of DetJ

Symbol of TrJ

stability

Symbol of DetJ

Symbol of TrJ

stability

Symbol of DetJ

Symbol of TrJ

stability

P1(0,0)

+

-

ESS

+

-

ESS

+

-

ESS

+

-

ESS

P2(1,0)

-

indeterminacy

Saddle Point

+

+

Unstable

-

indeterminacy

Saddle Point

+

+

Unstable

P3(0,1)

-

indeterminacy

Saddle Point

-

indeterminacy

Saddle Point

+

+

Unstable

+

+

Unstable

P4(1,1)

+

+

Unstable

-

indeterminacy

Saddle Point

-

indeterminacy

Saddle Point

+

-

ESS

P5(x**,y**)

-

0

Saddle Point

Case 1: At this time, the income of the owner supporting energy consumption data collection is less than that of not supporting energy consumption data collection, and the income of the ESCO choosing to provide energy consumption data collection and operation service is less than that of not providing energy consumption data collection and operation service.

Case 2: At this time, the income of the owner supporting energy consumption data collection is less than that of not supporting energy consumption data collection, and the income of the ESCO choosing to provide energy consumption data collection and operation service is greater than that of not providing energy consumption data col-lection and operation service.

Case 3: At this time, the income of the owner supporting energy consumption data collection is greater than that of not supporting energy consumption data collection, and the income of the ESCO choosing to provide energy consumption data collection and operation service is less than that of not providing energy consumption data col-lection and operation service.

Case 4: At this time, the income of the owner supporting energy consumption data collection is greater than that of not supporting energy consumption data collection, and the income of the ESCO choosing to provide energy consumption data collection and operation service is greater than that of not providing energy consumption data collection and operation service.

Conclusion: It can be seen from Table 4 that there is only one evolutionary equi-librium point (0, 0) in scenarios 1-3. The strategy of the system is that the owner does not support energy consumption data collection, and the ESCO does not provide ener-gy consumption data collection and operation services. In case 4, there are two evolu-tionary stable equilibrium points, namely (0,0) and (1,1). The corresponding (owner, ESCO) strategies are (do not support energy consumption data collection, do not pro-vide energy consumption data collection operation and maintenance services) and (support energy consumption data collection, provide energy consumption data col-lection operation and maintenance services). Only in this case, the system has the pos-sibility of evolving to (1,1) equilibrium point, but whether it can evolve to (1,1) equi-librium point depends on the initial state of the system. Therefore, the change of pa-rameters has a significant effect on the stability of the system.

Point 7: The conclusion is clear, but should be sustained by a stronger scientific content.

 Response 7: Thank you for your valuable comments. We find your opinion very useful. In response to this problem, we added the development of other countries in this field on page 11, and explained it in line 934/935 of the Conclusion, which confirmed the scientific nature of our research. The specific content we added is as follows:

In the United Kingdom, several changed policies have affected the actions and agency of actors, which also affect the transition towards low-energy buildings. Active energy efficiency policies promote the development of intermediaries, and existing organizations have adopted intermediary functions to advance low-energy homes in response to policy [60]. In the California region, building energy efficiency was achieved in the LAC (Los Angeles, California) through the means of state policy regulation. The results show the importance of designing energy efficiency policies for growing cities [61].

This result is corroborated by related papers from other countries, such as the impact of policy changes on building energy efficiency in the UK and California regions men-tioned on page 11.

We tried our best to improve the manuscript and made some changes marked in revised paper which will not influence the content and framework of the paper. We appreciate for Editor/ Reviews’ warm work earnestly, and hope the correction will meet with approval. Once again, thank you very much for you comments and suggestions.

Round 2

Reviewer 3 Report

Thanks to author for pointing the comments, the new version of the manuscript looks fine to be published.